# Quantity and quality: Normative open-access neuroimaging databases

**Scott Jie Shen Isherwood**[1]*, **Pierre-Louis Bazin**[1,2], **Anneke Alkemade**[1], **Birte Uta Forstmann**[1]

**1** Integrative Model-Based Cognitive Neuroscience Research Unit, University of Amsterdam, Amsterdam, The Netherlands, **2** Department of Neurophysics, Max Planck Institute for Human Cognitive and Brain Sciences, Leipzig, Germany

* scott@leeclan.net

## Abstract

The focus of this article is to compare twenty normative and open-access neuroimaging databases based on quantitative measures of image quality, namely, signal-to-noise (SNR) and contrast-to-noise ratios (CNR). We further the analysis through discussing to what extent these databases can be used for the visualization of deeper regions of the brain, such as the subcortex, as well as provide an overview of the types of inferences that can be drawn. A quantitative comparison of contrasts including T1-weighted (T1w) and T2-weighted (T2w) images are summarized, providing evidence for the benefit of ultra-high field MRI. Our analysis suggests a decline in SNR in the caudate nuclei with increasing age, in T1w, T2w, qT1 and qT2* contrasts, potentially indicative of complex structural age-dependent changes. A similar decline was found in the corpus callosum of the T1w, qT1 and qT2* contrasts, though this relationship is not as extensive as within the caudate nuclei. These declines were accompanied by a declining CNR over age in all image contrasts. A positive correlation was found between scan time and the estimated SNR as well as a negative correlation between scan time and spatial resolution. Image quality as well as the number and types of contrasts acquired by these databases are important factors to take into account when selecting structural data for reuse. This article highlights the opportunities and pitfalls associated with sampling existing databases, and provides a quantitative backing for their usage.

## Introduction

The purpose of this article is to summarize and compare some of the most prominent existing normative open-access structural magnetic resonance imaging (MRI) databases from a variety of research institutions, including our own Amsterdam ultra-high field adult lifespan (AHEAD) database [1]. The need for and benefit of open-access imaging databases has been emphasized in a number of recent reviews [2–4]. The community-wide movement towards open-access data sharing in the last decade is expected to massively advance the neuroimaging field and share the wealth of available data between researchers and institutions. Some of these

**Data Availability Statement:** The data used in this article are all from open-access databases and therefore all openly available to the public (please see S1 Table to find information on accessing specific databases). The code used in the sorting

and analysis is permanently uploaded to GitHub and publicly accessible via the following URL: https://github.com/scottyish/Quantity-and-Quality-Code.

**Funding:** The author(s) received no specific funding for this work.

**Competing interests:** The authors have declared that no competing interests exist.

benefits are obvious, such as the financial advantage of data sharing and the reuse of data between institutions. The re-analysis of acquired MR images also serves to aid reproducible research and provide multi-party levels of quality control. On top of this, the ability of new processing pipeline tools and analyses methods benefit greatly from the larger sum of data that the software can be trained on. It is important to acknowledge that large-scale data-sharing comes with its own disadvantages. Analyses based on post-hoc hypotheses and 'data-mining' can lead to spurious false positive findings [5]. Due to the sheer number of possible analyses in larger databases this problem grows increasingly likely. Data acquired within a specific framework and collected with a specific purpose may affect the extent to which this data can be used for separate analyses [6]. The questions investigated by the numerous neuroimaging databases described in this paper are diverse, with some attempting to bridge the gap between genetic influences and brain structure and others looking at the impact of the environment on the development of the human brain [7–9]. To this end, there are already a multitude of findings and publications arising from the data made accessible through these databases [10–22].

To our knowledge, a systematic comparison of these data, in terms of image quality, has not yet been published. This information is invaluable for the users of such databases to determine what conclusions they can reliably draw from the wealth of information provided. Through this analysis, we aim to aid in the accurate and valid use of the vast imaging data researchers have at their fingertips. Additionally, though many of these databases contain functional (f) MRI data (both resting state and task-specific), we will focus instead on brain anatomy and the inferences that can be made from structural imaging techniques. For comparisons, we will use quantitative measures of image quality, namely the signal-to-noise ratios (SNRs) and contrast-to-noise ratios (CNRs) associated with the MR images provided by each database. In short, the SNR infers the propensity of an MR image to delineate brain structures and detect pathology [23]. By providing these estimates for each database, we are giving a quantitative measure of two dimensions; image quality (SNR) and contrast (CNR). An increase in these quantitative measures improve the qualitative ability of e.g., manual or automatic parcellation. The CNR gives a valuable inference on the ability to spatially resolve detail in an image. Therefore, using different databases with varying CNRs may result in different outcomes depending on the reason they are being used (e.g., segmentation, volumetric measurements, delineation of cortical folding). SNR inherently provides an estimate of the noise level in a structure or image and higher image quality is both quantitively and qualitatively useful. Of course, the SNR is often used as a trade-off parameter to gain improvements in another aspect of the imaging method such as resolution, scan time, field of view (FOV) and indirectly, sample size. For example, a database with a low CNR and a large sample size may not be pragmatic to use for the parcellation of subcortical nuclei but would provide accurate volumetric whole brain estimates of a population. Conversely, a database with a high CNR and small sample size may not be able to provide reliable information at a population level but may deliver an insight into the substructure of a single region. Thus, larger databases with vast and multimodal data of each individual have already provided population-level information on cortical arrangement as well as the impact of genetics and the environment on the human brain [20, 24–28] which would not be possible in smaller databases.

There are currently at least 71 whole-body 7T MRI scanners worldwide [29]. Given the number of articles now specifically comparing 3T and 7T imaging of neurological disease, it is evident that higher field strengths are beneficial to answer questions in both the cognitive and clinical neurosciences [30–34]. The signal-to-noise ratio (SNR) increases in an almost linear fashion with field strength [35, 36] giving the potential for both greater spatial resolution and a higher CNR. Some of the databases described here have taken advantage of this, but the cost of use of these higher field strengths and their limited availability make it challenging for many

large-scale studies or institutions without access. Thus, the trade-off between the quantity and quality of acquired MR data arises.

The gains from ultra-high field MRI (UHF MRI) are especially important when investigating deeper regions of the brain (e.g., subcortex). UHF MRI can provide reduced partial volume effects due to increased spatial resolution, allowing for the visualization of finer anatomical detail [37, 38]. Historically, the lack of signal and contrast within the deep brain is the reason for the only recent development of subcortical maps in vivo [39, 40]. UHF MRI and its accompanying increase in SNR and contrast capacity will aid in the understanding of the structure of these deeper structures. Around 93% of the grey matter nuclei within the subcortex, making up almost a quarter of the total human brain volume, are currently not represented in standard MRI atlases [41, 42]. Some subcortical structures can be delineated through the use of these atlases, such as parts of the striatum, but most are too small to be manually or automatically parcellated [43]. Iron-rich structures including regions constituting the basal ganglia are difficult to delineate on standard T1w scans [44, 45], but specialised sequences can take advantage of the larger $T2^*$ contrast differences at higher field strengths [46]. For example, the abundance of iron in the substantia nigra (SN) and subthalamic nucleus (STN) make it an ideal target for $T2^*$ and SWI contrasts which can take advantage of these differences [47–50]. The delineation of these structures is made even harder by the limited SNR, due to the larger distance from the head coils [51].

Methods to improve image quality in MRI are not only limited to increasing the field strength of the scanner. The gradient strength, radiofrequency coils and use of optimized sequences also have a marked effect on acquisition efficiency. One such example is the Connectome scanner, of which there are currently only three in the world, which benefits from gradient strengths 3–8 times that found in standard 3T scanners. As with field strength, this factor facilitates both an increase in spatial resolution and a reduction scan time. Though previous studies have indicated the advantage of non-standard sequences (e.g., $T2^*$, QSM, SWI), owing to their capacity to increase the number of observable structures and to observe smaller brain regions (e.g., fibre tracks, nuclei) in deeper areas of the brain [48, 52]. The vast majority of databases focus on more standard T1w and T2w images, which are essential for volumetric calculations and distinguishing between grey and white matter regions but do not have the ability to quantify or delineate smaller and adjacently located nuclei [53, 54].

## Methods

The purpose of this article was not to present and analyse an exhaustive list of all currently available open-access neuroimaging databases, but to provide quantitative measures and accessing instructions for some of the most notable ones that meet our criteria. Most of the databases were identified using a structural MRI database list kindly provided from a cited paper which can be accessed here: https://github.com/cMadan/openMorph [4]. Two of the databases were identified as they were associated with the authors of this article [1, 55] and a further two databases were identified from the literature [56, 57]. All data was freely accessed in November 2018 and downloaded using the accessing instructions in S1 Table.

The selection criteria of the databases presented in this article were based on three characteristics. Firstly, the databases had to be normative, that is, made up of individuals that were reported as healthy at the time of scanning with no clinical presentation of neurological, psychiatric, neurodegenerative or peripheral disease. Secondly, the databases had to be a collection of curated images, uni- or multi-modal, that were acquired to be of similar composition (based on sequences and/or sites) to that of other images in the database. The reason for this criterion is that we assess five subjects randomly from each database and therefore must be

sure that their quality reflects that of the rest of the database accurately. Thirdly, these databases are open-access to the extent that they are accessible to the worldwide scientific community completely free of charge and without access barriers. Such access barriers include, for example, memberships, a specific institutional position (e.g., professorship) or the requirement of some type of institutional infrastructure (e.g., Federalwide Assurance).

The quality of the images acquired through the use of MRI are characterized by three main components: the acquired spatial resolution, the signal-to-noise ratio (SNR) and the contrast-to-noise ratio (CNR). These three aspects are in turn governed by the specific acquisition parameters used when obtaining the MR images. In the analysis presented here, the SNR was calculated by measuring the mean signal at the most posterior part of the corpus callosum (CC) and dividing it by its standard deviation. We also calculated the SNR of a grey matter region, namely the caudate nuclei (CN). To provide a measurement of CNR for each image, we computed the ratio of the difference in signal to the difference in noise of the CC and left and right caudate nuclei (LCN and RCN, respectively). These regions were chosen as we opted to compare the signal between a white matter area (CC) and a grey matter area (CN) of deeply situated brain regions. The SNR was calculated in both the left and right CN as a quality control step, under the assumption that these would yield similar SNR estimates. To test this, we used the programming language R and the 'BayesFactor' package to compute both frequentist and Bayesian t tests, respectively [58, 59]. The latter allows us to provide evidence for the null hypothesis (that there is no difference in signal between the left and right caudate nucleus). In order to have a singular SNR measure for both CN, we used the summation of the signal from 27 voxels from both regions and divided it by the standard deviation of the overall 54 voxels. This results in an SNR that is different than simply taking the mean of both SNR measurements for each CN. Eq 1 shows the calculation for which CNR values were computed. $\mu_{CC}$ indicates the mean signal of the CC, $\mu_{CN}$ indicates the mean signal of both CN. $\sigma_{CC}$ specifies the standard deviation of the CC and $\sigma_{CN}$ specifies the standard deviation of both CN.

$$CNR = \frac{(\mu_{CC} - \mu_{CN})}{\sqrt{\sigma_{CC}{}^2 + \sigma_{CN}{}^2}} \tag{1}$$

As many of the databases described here do not report SNR estimates, we decided to download a sample of the available data from each database and compute these indices to be able to make accurate comparisons between them. Importantly, even when SNR estimations were calculated by the databases, we performed a re-estimation to ensure that all SNRs presented here were estimated using the same protocol. The SNRs can be estimated using different structures and therefore derive values that are not always comparable between different procedures. To do this, five subjects from each database were selected at random and their available images downloaded. The SNRs and CNRs were calculated for each available contrast within each database. Databases that include large age ranges were split into age groups of young (18–28), middle-aged (34–53) or elderly (63–86). For these databases, five participants were taken from each age group so that we could compare SNR and CNR estimations across age-ranges. Although the proportion of each database used for the analysis may differ, they are statistically comparable as the same number of participants were selected randomly from the sample of each database.

For comparisons between databases the analysis focuses on the SNR of the CC ($SNR_{CC}$), for simplicity, unless otherwise stated. Sequences incorporating multiple echo times (e.g., MP2RAGEME) technically provide multiple contrasts in one sequence and were therefore all included in the estimates. For example, a sequence with four TEs (e.g., MP2RAGEME) would give four contrasts per participant. We chose five subjects to ensure feasibility of the manual

measures while accounting for potential variations in quality within a database. The CC and CN may not be the optimal structures for SNR comparisons for all of the contrasts for each database, but using these structures allows comparability over the entire analysis.

To calculate the SNR of the CC, LCN and RCN, one expert rater manually delineated the regions using the MIPAV imaging software (Medical Image Processing, Analysis and Visualization; [60]). Once the centre of the regions of interest were accurately delineated, a 3 x 3 x 3 cube of voxels was taken around one midpoint voxel to calculate the SNR using the mean and standard deviation of 27 contiguous voxels in the structure (Fig 1). A second method to calculate the SNR of these structures was also explored. Instead of taking 27 contiguous voxels, we took the voxel volumes of each image into account and measured the signal of 27 voxels from the same volumetric space. This involves simply normalizing the size of the cube of voxels measured by the images with the largest voxel size, so to measure from approximately the same area of each structure. The results were in line with what is reported here, and therefore we only report the measurements acquired from the first method.

To analyze the relationship between scan time and spatial resolution and scan time and $SNR_{CC}$, two linear regression models were setup. Both models used scan time as a predictor variable and either spatial resolution or $SNR_{CC}$ as the independent variable. This would allow us to observe a linear relationship between either of the two parameters. To correct for multiple comparisons, a Bonferroni adjustment was employed to maintain a 95% confidence in the analysis, giving a new significance threshold of 0.025.

We also present comparisons between slab images (small FOV) and whole brain images from databases that offer both, in order to demonstrate differences in SNR and CNR at higher resolutions. Both frequentist and Bayesian t tests were employed in R to compare these.

To compare age differences across multiple contrasts, linear mixed effect models from the 'lme4' R package were used [61]. Model 1 (null model) included the respective databases as a random intercept without adding any effect of age on SNR/CNR. Model 2 (full model)

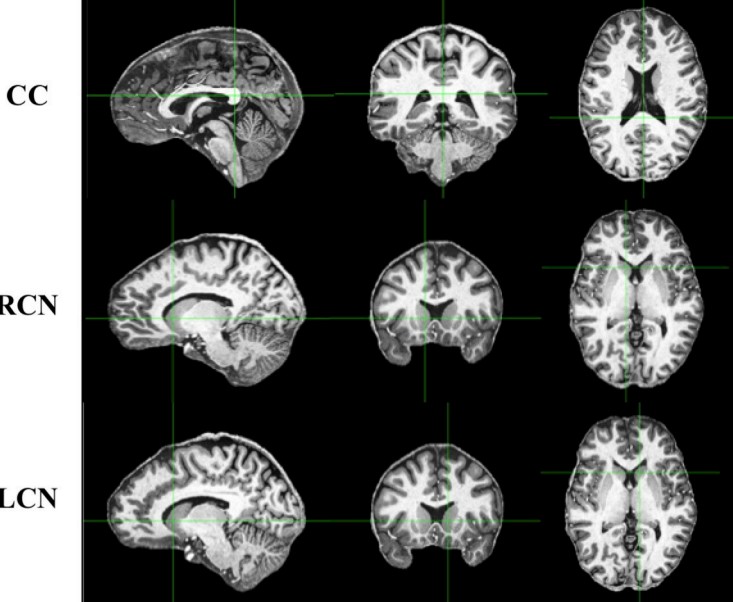

**Fig 1.** Sagittal (left), axial (middle), and coronal (right) views indicating the structures from which SNR measurements were taken. These T1w images were taken from one subject in the MPI-CBS database. CC, corpus callosum; RCN, right caudate nucleus; LCN, left caudate nucleus.

included both the database as a random intercept and age as a fixed effect. The likelihood estimations of each model were then compared by a likelihood ratio test though the use of an Analysis of Variance (ANOVA). A Bayesian linear modelling technique was also used, where the resultant Bayes factors were compared between model 1 and model 2. We opted to include the SNR and CNR data from all of the databases, even those without large age ranges, so to use as much of the wealth of information as possible for our statistical tests. This results in a larger centre of mass on the younger age group than the middle-aged and elderly groups, and although this does not result in an increase in power, it provides a more accurate estimate of the effect of age on SNR and CNR. For the model comparisons, age was used as a continuous predictor and therefore categorical ages were not used within the statistical analysis, these were only used for visualization.

To address the issue of reliability when taking a small subpopulation from large samples, we re-ran some of the SNR and CNR analysis with a different sample from the databases. 5 or 15 (if they included large age-ranges) additional samples were taken from each database that allowed it (dependent on the original sample size) and SNR measurements were calculated again from their T1w images for comparison against the original sample. Of the 20 databases included in this article, 17 had a sample size large enough for us to take additional measurements. SNR measurements were taken from the left caudate nucleus, right caudate nucleus and corpus callosum of 164 separate T1w images.

## Results

Based on our search, 41 databases were initially identified. After the first screening, 5 were excluded on the basis of access requirements. Of the remaining 36 databases, 20 were included in this article for description and comparison (see Fig 2 for Preferred Reporting Items for Systematic Reviews and Meta-Analyses; PRISMA flow diagram). Below we discuss these 20 databases that follow the three criteria including 250 [62], a completed Germany-based database which highlights its potential use for building an in vivo MR brain atlas due to its ultrahigh resolution whole brain images of one subject; Age-ility [63], a completed Australia-based database investigating the relationship between cognitive control and adaptive/maladaptive behaviours across the adult lifespan; the AHEAD database [1], an ongoing Netherlands-based database aiming to acquire high-resolution images of the human subcortex and map so-called *terra incognita*; the Atlasing of the Basal Ganglia (ATAG) project [55], a completed Netherlands/ Germany-based database whose aim was to acquire high-resolution data to observe anatomical differences over the adult lifespan; the Brain Genomics Superstruct Project (GSP; [9]), a completed US-based database looking to solidify and find links between brain function, behaviour and genetic variation; the Cambridge Centre for Aging and Neuroscience (Cam-Can; [64, 65]), an ongoing UK-based database aiming to characterize age-related changes in cognition and brain structure and function, and to uncover the neurocognitive mechanisms that support healthy aging; the Dallas Lifespan Brain Study (DLBS; http://fcon_1000.projects.nitrc.org/indi/ retro/dlbs.html), an ongoing US-based database designed to accelerate our understanding of both the preservation and decline of cognitive functioning across the adult lifespan; the Human Connectome Project Young Adult (HCP-YA; [8, 66, 67]), an ongoing US-based database aiming to generate a complete and accurate description of the connections amongst grey matter locations in the human brain at the millimeter scale. Information eXtraction from Images (IXI; http://www.brain-development.org), a completed UK-based database from three London Hospitals aimed to aid in decision support in healthcare and the analysis of images obtained in drug discovery; Kirby 21 [68], a completed US-based database aiming to assess the scan-rescan reproducibility of a 60 minute scanning session, wanting to establish a baseline for

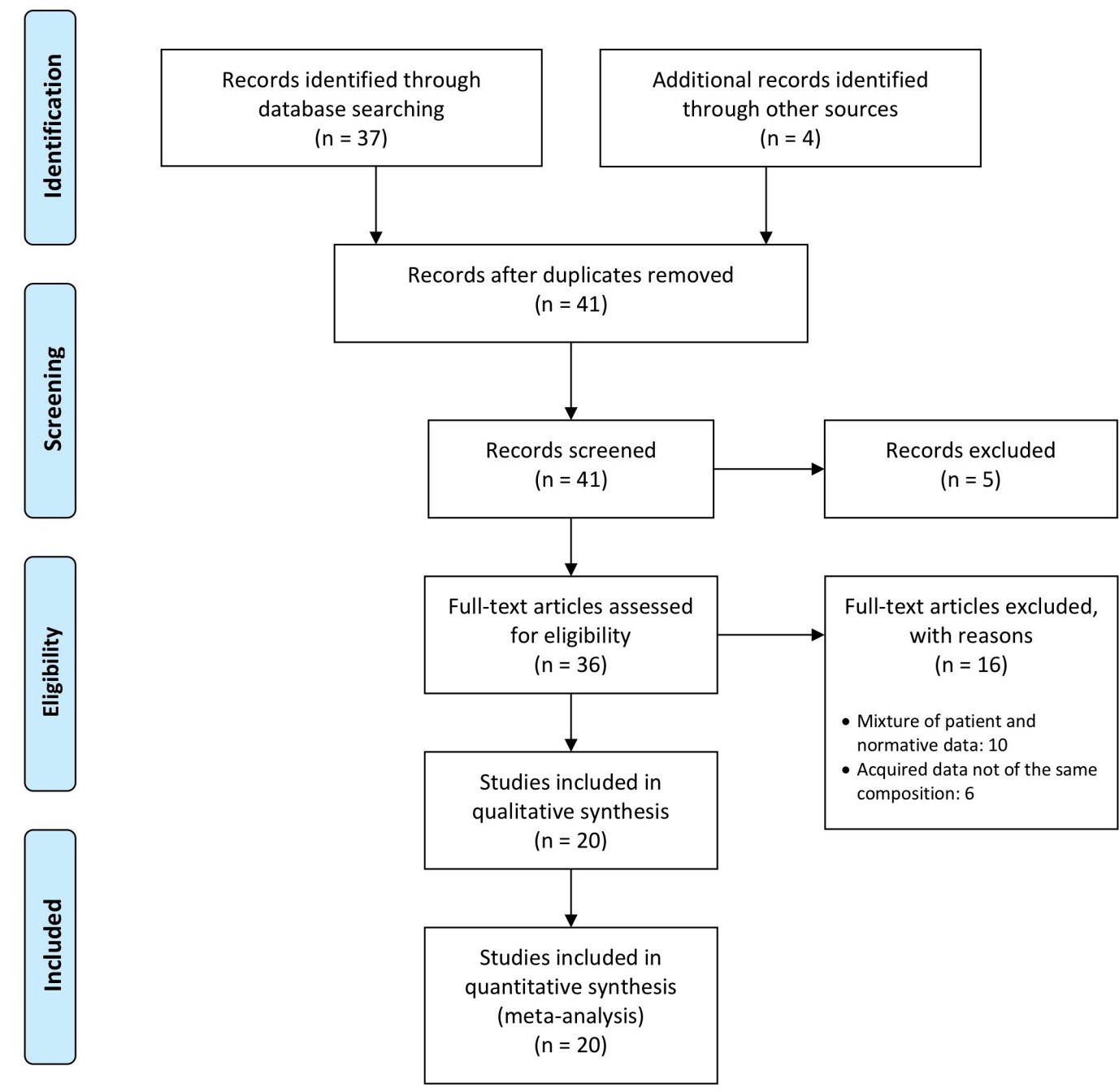

*From:* Moher D, Liberati A, Tetzlaff J, Altman DG, The PRISMA Group (2009). *P*referred *R*eporting *I*tems for *S*ystematic Reviews and *M*eta-*A*nalyses: The PRISMA Statement. PLoS Med 6(7): e1000097. doi:10.1371/journal.pmed1000097

For more information, visit **www.prisma-statement.org**.

**Fig 2. PRISMA flow diagram.**

developing multi-parametric imaging protocols; Maastricht [69], a completed Netherlands-based database with the aim of facilitating the development of segmentation algorithms on the challenging nature of 7T MR data; Multiple Acquisitions for Standardization of Structural Imaging Validation and Evaluation (MASSIVE; [70]), a completed Netherlands-based single-subject dataset aiming to serve as a representative testbed for diffusion-MRI correction strategies, image processing techniques and microstructural modelling approaches; the Midnight Scan Club (MSC; [71]), a completed US-based database of scientific volunteers wanting to increase our understanding of brain function on the individual level, as opposed to just the central tendencies of populations; the Max Planck Institute–Human Brain and Cognitive Sciences repository (MPI-CBS; [57]), a completed Germany-based database wanting to stimulate the development of imaging processing tools for high resolution and quantitative imaging, that have been mainly designed for lower quality images; Max Planck Institute–Leipzig Mind Brain Body (MPI-LMBB; [72]), another completed Germany-based databases which aimed to explore individuals variance across cognitive and emotional phenotypes in relation to the brain; Nathan Kline Institute–Rockland Sample (NKI-RS; [7]), an ongoing US-based database aiming to provide normative trajectories of brain development so to facilitate the identity of pathological markers; Pediatric Template of Brain Perfusion (PTBP; [73]), a completed US-based database focusing on increasing our understanding of adolescent brain development with multi-model MR imaging and its relationship with both environmental and cognitive measures; RAIDERS [56], a completed US-based database focusing on functional imaging during segments of full-length feature film "Raiders of the Lost Ark"; the Southwest University Adult Lifespan Dataset (SALD; [74]), a completed China-based database aiming to observe how the normative brain changes structurally and functionally over the adult lifespan; and StudyForrest [75, 76], an ongoing German-based database aiming to provide data in a more complex setting, as opposed to the simplified experimental designs normally used, to therefore provide a more ecologically valid insight into brain function.

Table 1 presents an overview of these databases including information on field strength, sequences and the number of participants. Example T1-weighted (T1w) images taken from each database are presented in Fig 3. Further information, including the website address and accessing instructions of each database can be found in S1 Table. Detailed descriptions of the individual databases can be found on their website address or descriptor papers.

We would like to acknowledge the importance of other neuroimaging databases that do not meet our selection criteria, such as the Open Access Series of Imaging Studies (OASIS; [77, 78]), 1000 Functional Connectome Project (FCP; [79]), Alzheimer's Disease Neuroimaging Initiative (ADNI; [80, 81]), Autism Brain Imaging Data Exchange (ABIDE; [82]), Brain Images of Normal Subjects (BRAINS; [83]), Australian Imaging Biomarkers and Lifestyle Study of Aging (AIBL; [84]), Pediatric Imaging, Neurocognition, and Genetics (PING; [85]), Adolescent Brain Cognitive Development (ABCD) study [86], Attention Deficit Hyperactivity Disorder (ADHD) 200 [87], Child Mind Institute Healthy Brain Network (CMI-HBN; [88]), Center for Biomedical Research Excellence (COBRE; http://fcon_1000.projects.nitrc.org/indi/retro/cobre.html), Consortium for Reliability and Reproducibility (CoRR; [89]), Function Biomedical Informatics Research Network (fBIRN; [90]), Minimal Interval Resonance Imaging in Alzheimer's Disease (MIRIAD; [91]), National Alzheimer's Coordinating Center (NACC; [92]), National Consortium on Alcohol and Neurodevelopment in Adolescence (NCANDA; [93]), Philadelphia Neurodevelopmental Cohort (PNC; [94]), Mindboggle-101 [95], SchizConnect [96], OpenNeuro [97] and the UK Biobank [98]. These databases, such as the ABCD database, and the PING database are also of great interest, but they are not openly available to researchers outside of NIH institutions, and thus do not meet our criteria for inclusion in this study (see S2 Table for an overview of the inclusion criterion these databases did not meet).

**Table 1. Overview of the acquisition parameters and details of each database.**

| Database | N | Field strength (T) | Structural sequence | Contrast | Slice thickness (mm) | TR (ms) | TE (ms) | TI (ms) | Flip angle (deg) | Bandwidth (Hz/px) | Scan time (min:sec) | Voxel size (mm) |
|---|---|---|---|---|---|---|---|---|---|---|---|---|
| 250 | 1 | 7 | MPRAGE | T1w | 0.25 | 3580 | 2.41 | 1210 | 5 | 440 | 53:00 | 0.25 x 0.25 x 0.25 |
| AHEAD | 106 | 7 | MP2RAGEME | T1w, T2*w, PDw, QSM | 0.7 | $6.2_{TR1}$, $31_{TR2}$ $6778_{TR}$ | 3 | 670, 3675.4 | 4/4 | 404.9 | 19:53 x 2 | 0.64 x 0.64 x 0.7 |
|  |  |  | MP2RAGEME-sb | T1w, T2*w, PDw | 0.5 | $8_{TR1}$, $32_{TR2}$ $8330_{TR}$ | 3, 11.5, 19, 28.5 4.6 4.6, 12.6, 20.6, 28.6 | 670, 3738 | 7/8 | - | - | 0.5 x 0.5 x 0.5 |
| AGE-ILITY | 131 | 3 | MPRAGE | T1w | 1 | 2200 | 3.5 | - | 7 | 190 | 6:45 | 1 x 1 x 1 |
| ATAG | 53 | 7 | ME-3D-FLASH | T2*w | 0.5 | 41 | 11.22, 20.39, 29.57 | | 14 | 160 | 17:18 | 0.5 x 0.5 x 0.5 |
|  |  |  | MP2RAGE | T1w | 0.7 | 5000 | 2.45 | 900, 2750 | 5/3 | 250 | 10:57 | 0.7 x 0.7 x 0.7 |
|  |  |  | MP2RAGE-sb | T1w | 0.6 | 5000 | 3.71 | 900, 2750 | 5/3 | 240 | 9:07 | 0.6 x 0.6 x 0.6 |
| GSP | 1570 | 3 | MEMPRAGE | T1w | 1.2 | 2200 | 1.5, 3.4, 5.2, 7.0 | 1100 | 7 | 651 | 2:12 | 1.2 x 1.2 x 1.2 |
| CAM-CAN | 280 | 3 | MPRAGE | T1w | 1 | 2250 | 2.99 | 900 | 9 | - | 4:32 | 1 x 1 x 1 |
|  |  |  | SPACE | T2w | 1 | 2800 | 408 | - | 9 | - | 4:30 | 1 x 1 x 1 |
| DLBS | 315 | 3 | MPRAGE | T1w | 1 | 8135 | 3.7 | 1100 | 12 | | 3:57 | 1 x 1 x 1 |
| HCP-YA | 1206 | 3 | 3D MPRAGE | T1w | 07 | 2400 | 2.14 | 1000 | 8 | 210 | 7:40 | 0.7 x 0.7 x 0.7 |
|  |  |  | 3D SPACE | T2w | 0.7 | 3200 | 565 | - | Variable | 744 | 8:24 | 0.7 x 0.7 x 0.7 |
| IXI | 619 | 3 | - | T1w | 1.2 | 9.6 | 4.6 | - | 8 | - | - | 0.94 x 0.94 x 1.2 |
|  |  |  | - | T2w | 1.2 | 5725.79 | 100 | - | 90 | - | - | 0.9 x 0.9 x 1.2 |
|  |  |  | - | PDw | 1.2 | - | - | - | - | - | - | 0.9 x 0.9 x 1.2 |
| KIRBY 21 | 21 | 3 | MPRAGE | T1w | 1.2 | 6.7 | 3.1 | 842 | 8 | - | 5:56 | 1 x 1 x 1.2 |
|  |  |  | FLAIR | T2w | 1.1 | 8000 | 330 | 2400 | - | - | 8:48 | 1.1 x 1.1 x 1.1 |
| MAASTRICHT | 5 | 7 | MPRAGE | T1w | 0.7 | 3100 | 2.42 | 1500 | 5 | 182 | 13:30 | 0.7 x 0.7 x 0.7 |
|  |  |  | MPRAGE | PDw | 0.7 | 1380 | 2.42 | - | 5 | 182 | 5:53 | 0.7 x 0.7 x 0.7 |
|  |  |  | MPRAGE | T2*w | 0.7 | 4910 | 16 | - | 5 | 473 | 21:20 | 0.7 x 0.7 x 0.7 |
| MASSIVE | 1 | 3 | 3D-IR-TSE | FLAIR | 1 | 4800 | 300 | 1650 | 90 | - | 3:45 | 1 x 1 x 1 |
|  |  |  | 3D-TFE | T1w | 1 | 8000 | 1.25 | - | 8 | - | 3:45 | 1 x 1 x 1 |
|  |  |  | 3D-TSE | T2w | 1 | 2500 | 213 | - | 90 | - | 2:45 | 1 x 1 x 1 |
| MSC | 10 | 3 | - | T1w | 0.8 | 2400 | 3.7 | 1000 | 8 | - | - | 0.8 x 0.8 x 0.8 |
|  |  |  | - | T2w | 0.8 | 3200 | 497 | - | - | - | - | 0.8 x 0.8 x 0.8 |
| MPI-CBS | 28 | 7 | MP2RAGE | T1w | 0.5 | 5000 | 2450 | 900, 2750 | 5/3 | 250 | 28:02 | 0.5 x 0.5 x 0.5 |
|  |  |  | ME-FLASH | T2*w | 0.5 | 44 | 9.18, 17.33, 25.49, 33.65 | - | 14 | 200 | 26.12 | 0.5 x 0.5 x 0.5 |
| MPI-LMBB | 321 | 3 | 3D-MP2RAGE | T1w | 1 | 5000 | 2.92 | 700, 2500 | 4/5 | 240 | 8:22 | 1 x 1 x 1 |
|  |  |  | FLAIR | - | 1 | - | - | - | - | - | - | 0.49 x 0.49 x 1 |
| NKI-RS | 1000 | 3 | MPRAGE | T1w | 1 | 1900 | 2.52 | 900 | 9 | 170 | 4:18 | 1 x 1 x 1 |
| PTBP | 120 | 3 | MPRAGE | T1w | 1 | 2170 | 4.33 | 1100 | 7 | - | 8:08 | 1 x 1 x 1 |
| RAIDERS | 11 | 3 | MPRAGE | T1w | 1 | 9850 | 4530 | - | 8 | - | - | 0.938 x 0.938 x 1 |
| SALD | 494 | 3 | MPRAGE | T1w | 1 | 1900 | 2.52 | 900 | 9 | 170 | 4:26 | 1 x 1 x 1 |
| STUDYFORREST | 20 | 3 | 3D-TFE | T1w | 0.7 | 2500 | 5.7 | 900 | 8 | 144 | 12:49 | 0.67 x 0.67 x 0.7 |
|  |  |  | 3D-TSE | T2w | 0.7 | 2500 | 230 | - | - | 744.8 | 7:40 | 0.67 x 0.67 x 0.7 |
|  |  |  | 3D Presto FFE | SWI | 0.35 | 19 | 26 | - | 10 | 217.2 | 7:13 | 0.43 x 0.43 x 0.35 |

N, number of participants; TR, repetition time; TE, Echo time; TI, inversion time; MPRAGE, magnetization prepared rapid gradient echo; MP2RAGE, magnetization prepared 2 rapid acquisition gradient echoes; ME, multiple echo; FLASH, fast low angle shot; SPACE, sampling perfection with application of optimized contrasts using different flip angle evolutions; FLAIR, fluid attenuation inversion recovery; IR, inversion recovery; TSE, turbo spin echo; TFE, turbo field echo; sb, slab;-indicates unknown or not applicable information.

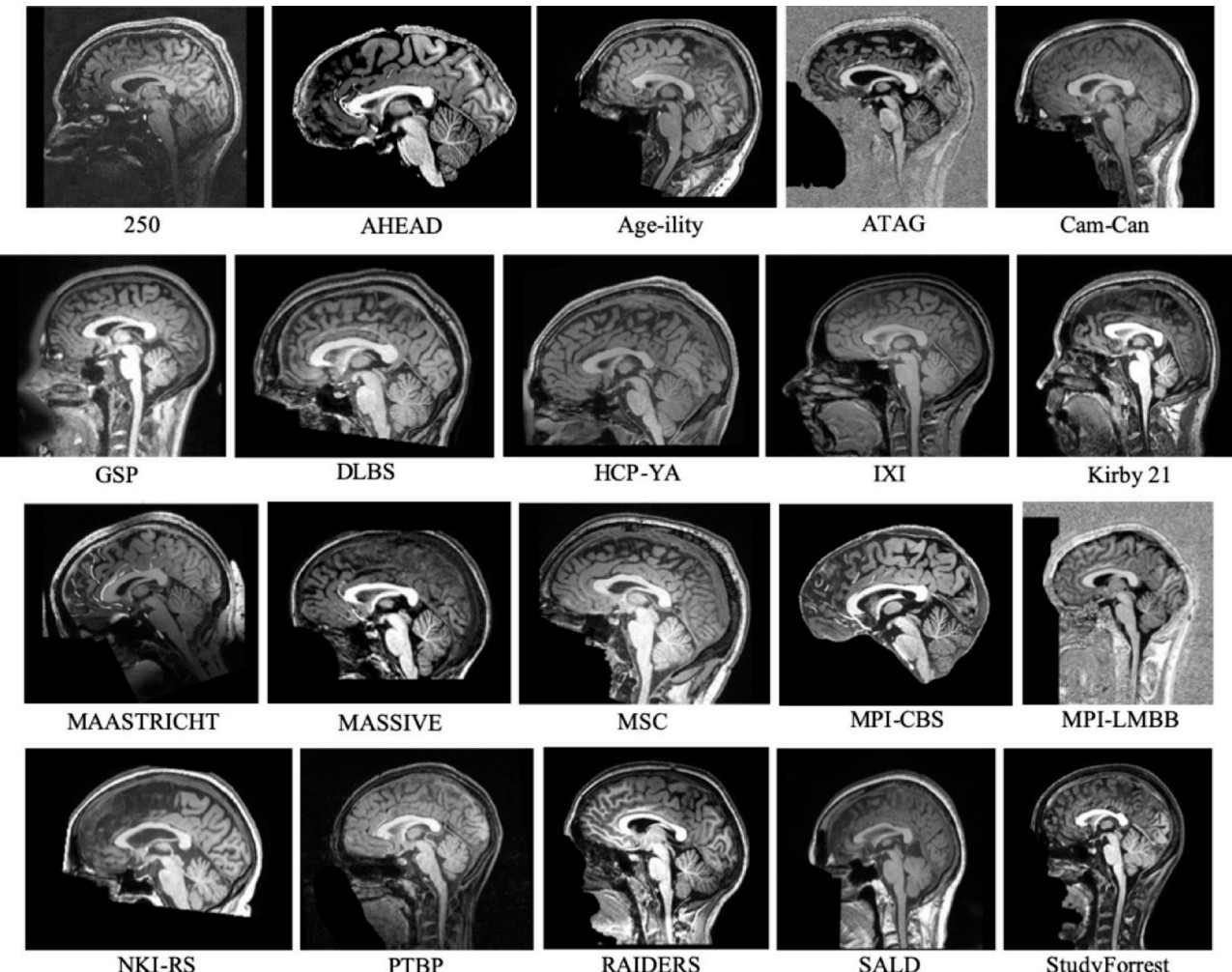

**Fig 3. Mid-sagittal T1w images from each neuroimage database.** One participant was selected at random from each of the databases to serve as an example of the image quality expected.

Additionally, we would like to recognize that many clinical databases also contain images of healthy individuals. The reuse of databases consisting of only healthy individuals is more convenient, creating an even lower threshold for the reuse of data. We would like to emphasize that the exclusion of normative data from clinical databases, databases containing non-harmonious data or databases that have institutional and/or positional requirements is in no way a comment on their data quality or usefulness.

Because of the large age-ranges, fifteen participants were used for the following subset of databases (AHEAD, ATAG, CAMCAN, DLBS, IXI, MPI-LMBB, NKI-RS, SALD). Ten databases therefore present a mean SNR value of five participants, eight databases present a mean SNR value of fifteen participants and two databases (MASSIVE and 250) were comprised of only one subject. For this case, five scanning sessions were taken, and the mean SNR calculated. 670 images were analysed in total for the main analysis, and a further 164 to test the reliability of the initial sample.

Described below are the results of the SNR and CNR analysis. To comply with the Health Insurance Portability and Accountability Act (HIPAA, https://www.hhs.gov/hipaa/index.html)

and the European equivalent General Data Protection Regulation (GDPR, https://eugdpr.org/), it is agreed upon by the scientific community that high resolution MRI images give the means for identifiability and facial reconstruction and must therefore be subject to precautionary measures to ensure privacy [99]. Therefore, the images provided here by the cited databases are coupled with a defacing mask to protect against identifiability, with the exception of the IXI and MSC databases. Other than this essential step, all included databases offer unprocessed images or both unprocessed and pre-processed images, with the exception of the MPI-CBS database. When available, all calculations regarding SNR and CNR used the unprocessed MR images.

Due to the inherent trade-off between SNR and spatial resolution, we opted to normalize the SNR and CNR by dividing the original ratio values by the voxel dimensions of the acquired images. This gives a more accurate depiction of the image quality of each database. Therefore, unless otherwise specified, or in the case of quantitative images, we show normalized SNR values, not raw SNR values. A graphical comparison of the raw SNR and the normalized SNR for the T1w images of each database is shown in S1 Fig.

## Comparing T1w images

As all the databases presented here contained a T1w image for each participant, these were used as the main sample to be compared. A frequentist and Bayesian paired t-test were used to compare the values of the two caudate nuclei within each database, concluding that there were indeed no significant differences between the calculated SNR, as there is substantial evidence for the null hypothesis ($p = 0.630$, $t = 0.492$, $DF = 14$, $BF = 0.292$; [100]). The SNR of each nuclei, averaged by database, are visualized in Fig 4.

Fig 5 visualizes the relationship between sample size and SNR. This indirect tradeoff between the two is perhaps anticipated, simply due to the costs associated with both an increased number of participants and superior acquisition methods (e.g., higher field strengths and increased scan time). Of course, both sides of the spectrum are accompanied with their own advantages and disadvantages. Larger sample sizes can reduce the susceptibility to spurious findings and deliver greater statistical power, but may have to sacrifice some features of the imaging data (e.g., voxel resolution, SNR or number of modalities). For example, databases with large sample sizes and large voxel sizes may not be suitable for studying morphometric changes that occur in small subcortical nuclei but can provide accurate estimations of cortical thickness with a high statistical power.

Fig 6 displays the normalized $SNR_{CC}$ and CNR values of the T1w images from each database. The results are present as ascending from bottom to top, based on their SNR estimation, ranging from 15.8 (GSP) to 292.3 (250 database). Their numerical values are presented in Table 2.

To investigate the similarity of the first sample of measurements to the second sample, a Bayesian ANOVA was used to provide evidence for or against the null hypothesis (that these samples were taken from the same distribution). The Bayes Factors (BF) resulting from this analysis for each structure are as follows: corpus callosum SNR BF = 0.139, caudate nuclei SNR BF = 0.129, and the CNR BF = 0.224. Based on [100] this provides substantial evidence for the null hypothesis, that both samples from each database come from the same distribution. This shows that our sample-based method is reproducible across samples of the databases. Although it would ideally be best to manually segment the CC and CN in each subject, the simplified approach we take here provides a good trade-off between accuracy and manageability given the large of amount of manual delineation that had to be done in the original (665) and the second (164) sample.

## SNR estimations for left and right caudate nucleus

**Fig 4. SNR estimations for the left and right caudate nucleus.** Data averaged over the individuals of each database. Error bars indicate standard error of the mean.

### Comparing T2w images

Fig 7 presents an overview of the estimated $SNR_{CC}$ and CNR of the six databases containing T2w images. The results are ordered as ascending from bottom to top, based on the $SNR_{CC}$ estimation, ranging from 11.8 (Cam-Can) to 51.6 (StudyForrest). Their numerical values can be found in Table 3.

### Relationships with scan time

We then turned to analyze the relationships between scan time and $SNR_{CC}$ as well as scan time and the acquired spatial resolution. Fig 8A shows the relationship between scan time and the normalized SNR for both 3T and 7T scanners separately. It can be seen that there is a significant positive correlation within the 3T data, and the 7T data displays the same trend but does not show significance. This relationship is expected, since longer scan times are associated with better image quality. In addition to scan time predicting image quality in terms of $SNR_{CC}$, a negative relationship between scan time and the acquired voxel volume was found (Fig 8B). Longer scan times in the presented databases are therefore indicative of better T1w images both in terms of SNR and spatial resolution.

### Quantitative T1 and FOV

Four databases provide quantitative T1 maps (qT1) in addition to T1w images. Two of these databases also provide both whole-brain images as well as slabs with higher resolution and a

## Sample size vs SNR

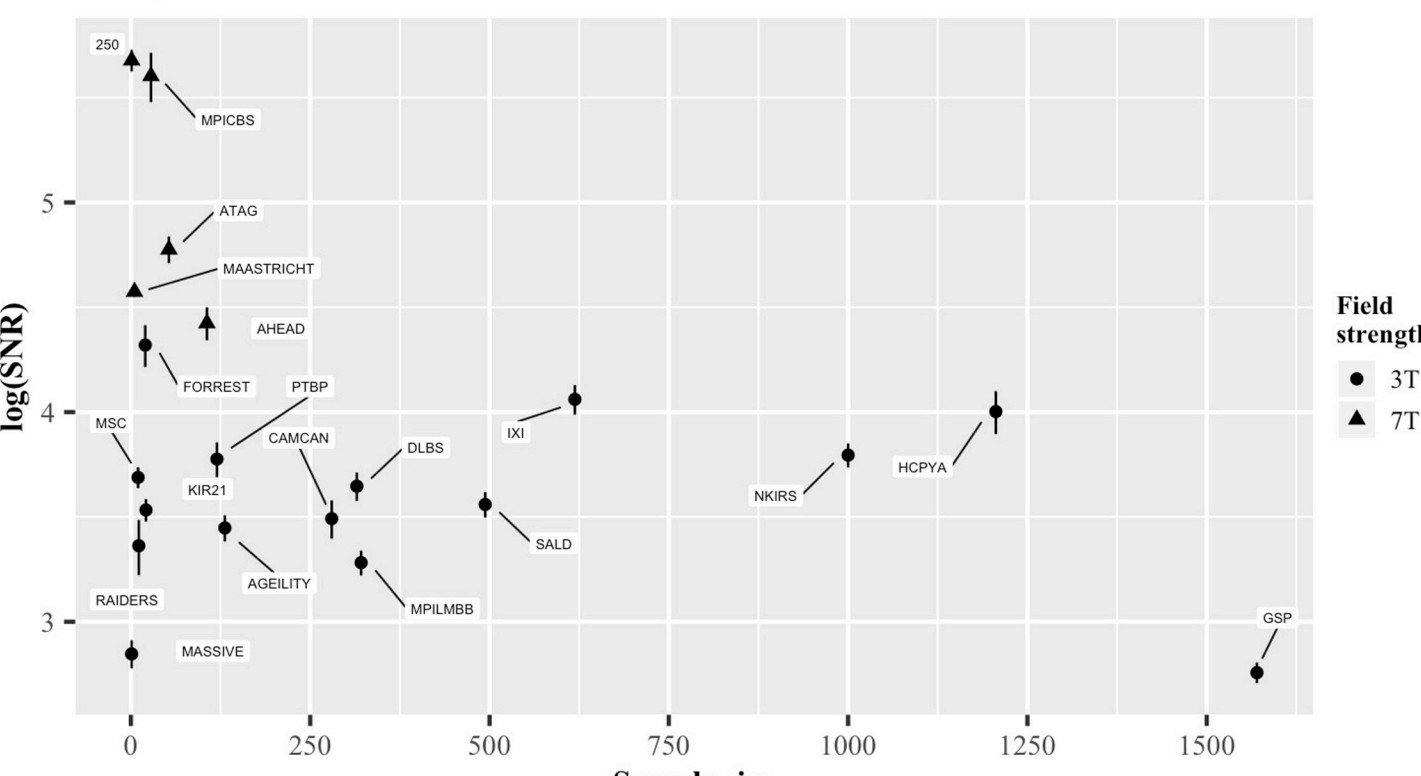

**Fig 5. The relationship between sample size and SNR$_{CC}$.** Error bars indicate standard error of the mean. Both the SNR values and the standard errors are presented on the log scale. Circular symbols indicate 3T data, triangular symbols indicate 7T data.

smaller FOV. A comparison of the normalized SNR$_{CC}$ associated with the qT1 and T1w images of the same databases are shown in Fig 9.

Table 4 displays the SNR$_{CC}$ and CNR associated with the whole-brain and slab images of the same contrasts acquired by these two databases (ATAG and AHEAD). A frequentist and Bayesian paired t-test indicates that the slab images have a significantly larger SNR$_{CC}$ than the whole brain images, demonstrating the benefits of high resolution (p = 0.0017, t = 6.06, DF = 5, BF = 25.81). Though, this does not appear to translate to a higher CNR (p = 0.074, t = 2.3, DF = 5, BF = 1.57).

### Age-related differences

Figs 10 and 11 display the differences in the SNR and CNR across the age groups of young (age: 18–28), middle-aged (age: 34–53) and elderly subjects (age: 63–86) in both T1w and T2w images. 165 T1w images and 50 T2w images were used for model comparison. For the SNR$_{CC}$ on the T1w images, the full model comprising age as a predictor was a significantly better fit than the null model (p = 0.011). This relationship was also found for the T1w CNR results (p = 0.00037). In addition to age-related differences in the SNR of white matter areas (SNR$_{CC}$) and the CNR of T1w images, we also tested the relationship between age and the SNR of a grey matter region (SNR$_{CN}$). A significant effect of age was found, indicating a loss of signal in the CN over age (p = 0.0062). We then turned to analyze the effect of age on the MR signal of T2w images. Again, we were interested in age differences in the SNR$_{CC}$ the SNR$_{CN}$ and the contrast

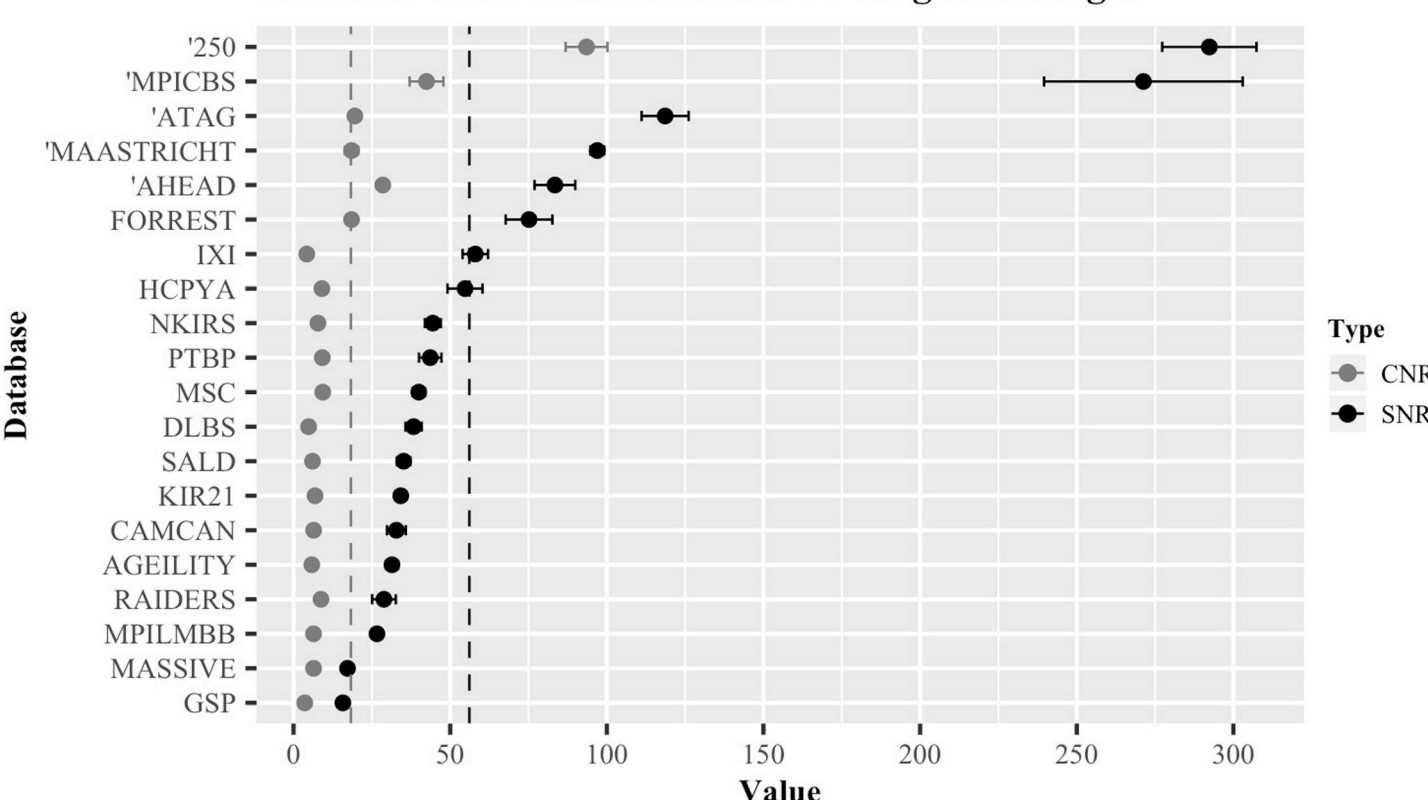

**Fig 6. Overview of ratios for databases containing a T1w image.** $SNR_{CC}$ values are shown in black, CNR values in grey. Each value has been normalized by the voxel dimensions specific to the image it describes. Error bars indicate standard error of the mean. Databases marked with an apostrophe (') indicate 7T data. The dotted vertical lines indicate the mean of the $SNR_{CC}$ (black) and CNR (grey).

difference between the grey and white matter regions (CNR). Similarly to the T1w images, an age-related decline in $SNR_{CN}$ and CNR was observed in the T2w images (p = 0.0019, p = 0.000022, respectively) even though the model comparison indicated that the age-related differences in $SNR_{CC}$ were non-significant in the T2w images (p = 0.24).

To gain a greater insight into the relationship between age and the acquired signal from these white and grey matter structures, we used a Bayesian linear modelling technique. The resulting BFs from this method indicated a less conclusive relationship than its frequentist counterpart in some respects. In terms of an age-related reduction in signal within the T1w images, moderate evidence was found for this hypothesis in the CN (BF = 5.52), followed by further moderate evidence within the CC (BF = 4.29), and very strong evidence for this hypothesis was found for the CNR (BF = 61.90). Turning to the T2w images, no evidence in either direction was found for an age-related reduction in signal from the CC (BF = 1.89), strong evidence was found for this hypothesis in the CN (BF = 14.51), and across the age groups, the CNR appeared to show extreme evidence for a relationship (BF = 504.22). Taken together, these results suggest the presence of an age-related deterioration in signal in the caudate nuclei, inferred by both the T1w and T2w images.

As a further assessment of age-related differences, we also compared the SNR and CNR values across qT1 and qT2* images. We again compared linear mixed effect models including age as a fixed effect and the database as a random intercept to a null model without an effect of

**Table 2. Summary table describing the SNR$_{CC}$, SNR$_{CN}$ and CNR of the T1w images of each database.**

| Database | Sequence | Contrast | SNR$_{CC}$ | SNR$_{CN}$ | CNR | n |
|---|---|---|---|---|---|---|
| 250 | MPRAGE | T1w | 292.3 ± 15.0 | 198.3 ± 14.6 | 93.5 ± 6.7 | 1 |
| AHEAD | MP2RAGEME | T1w | 83.4 ± 6.5 | 39.5 ± 1.3 | 28.5 ± 1.3 | 15 |
| Age-ility | MPRAGE | T1w | 31.4 ± 2.0 | 20.4 ± 2.0 | 5.8 ± 0.9 | 5 |
| ATAG | MP2RAGE | T1w | 118.6 ± 7.5 | 29.6 ± 1.4 | 19.6 ± 0.7 | 15 |
| Cam-Can | MPRAGE | T1w | 32.8 ± 3.0 | 24.1 ± 1.2 | 6.1 ± 0.5 | 15 |
| GSP | MEMPRAGE | T1w | 15.8 ± 0.8 | 8.8 ± 0.7 | 3.6 ± 0.2 | 5 |
| DLBS | MPRAGE | T1w | 38.3 ± 4.2 | 19.4 ± 2.7 | 4.8 ± 0.7 | 15 |
| HCP-YA | MPRAGE | T1w | 54.7 ± 5.6 | 40.4 ± 1.8 | 9.1 ± 1.3 | 5 |
| IXI | - | T1w | 58.0 ± 4.1 | 33.7 ± 1.8 | 4.2 ± 0.4 | 15 |
| Kirby 21 | MPRAGE | T1w | 34.2 ± 1.8 | 16.7 ± 1.1 | 6.9 ± 0.6 | 5 |
| MAASTRICHT | MPRAGE | T1w | 96.9 ± 2.2 | 36.6 ± 3.3 | 18.5 ± 2.1 | 5 |
| MASSIVE | 3DTFE | T1w | 17.2 ± 1.2 | 10.6 ± 0.3 | 6.4 ± 0.3 | 1 |
| MSC | - | T1w | 40.0 ± 2.0 | 26.6 ± 1.3 | 9.3 ± 0.2 | 5 |
| MPI-CBS | MP2RAGE | T1w | 271.3 ± 31.8 | 93.1 ± 15.9 | 42.4 ± 5.4 | 5 |
| MPI-LMBB | MP2RAGE | T1w | 26.6 ± 1.6 | 12.7 ± 0.3 | 6.4 ± 0.3 | 15 |
| NKI-RS | MPRAGE | T1w | 44.5 ± 2.5 | 28.1 ± 1.0 | 7.8 ± 0.4 | 15 |
| PTBP | MPRAGE | T1w | 43.6 ± 3.6 | 28.2 ± 1.8 | 9.2 ± 0.7 | 5 |
| RAIDERS | MPRAGE | T1w | 28.9 ± 3.8 | 16.9 ± 1.6 | 8.8 ± 0.9 | 5 |
| SALD | MPRAGE | T1w | 35.1 ± 2.1 | 23.9 ± 0.8 | 6.0 ± 0.2 | 15 |
| StudyForrest | 3DTFE | T1w | 75.2 ± 7.5 | 43.5 ± 4.1 | 18.5 ± 1.1 | 5 |

Each ratio value is shown as the mean of all the subjects ± the standard error of the mean. n indicates the number of subjects used for the calculations.

age. One database, MPILMBB, provides age ranges of five years for each of their participants as opposed to a single age value, presumably for privacy purposes. In order to derive reliable estimates when comparing these mixed effect models, we randomly sampled ages for participants in this database from a uniform distribution of the age range reported. We then iterated over this a total of 1000 times and calculated results from the frequentist and Bayesian model comparisons for each sampled age, below we report the mean results for these iterations. Similarly to the T1w and T2w comparisons, both qT1 and qT2* maps showed a significant change in CNR across the adult lifespan (p = 0.00032, BF = 73.25; p = 0.035, BF = 1.12). The SNR$_{CN}$ significantly declined in both the qT1 and qT2* images (p = 0.0015, BF = 19.68; p = 0.00019, BF = 247.99, respectively). A similar decline was found for the SNR$_{CC}$ in the qT2* images (p = 0.00063, BF = 34.83). Although, only a negligible decline in signal was found for the CC in the qT1 images (p = 0.020, BF = 2.49).

## Discussion

We present the first quantitative comparison exploring the image quality offered by twenty open-access databases of structural MRI freely available to researchers world-wide. To this end, SNRs were calculated from both the corpus callosum and caudate nuclei. From these calculations, CNRs were derived, which in this case can indicate the extent to which these images can distinguish between grey and white matter. An additional analysis assessed differences in T1w and T2w SNR values across the adult lifespan, taking advantage of larger imaging databases with accompanying demographic information and large age ranges. Due to the wealth of data provided by these databases, clear relationships between the scan time and both acquired voxel dimensions and acquired SNRs could also be found, indicating the efficiency of specific

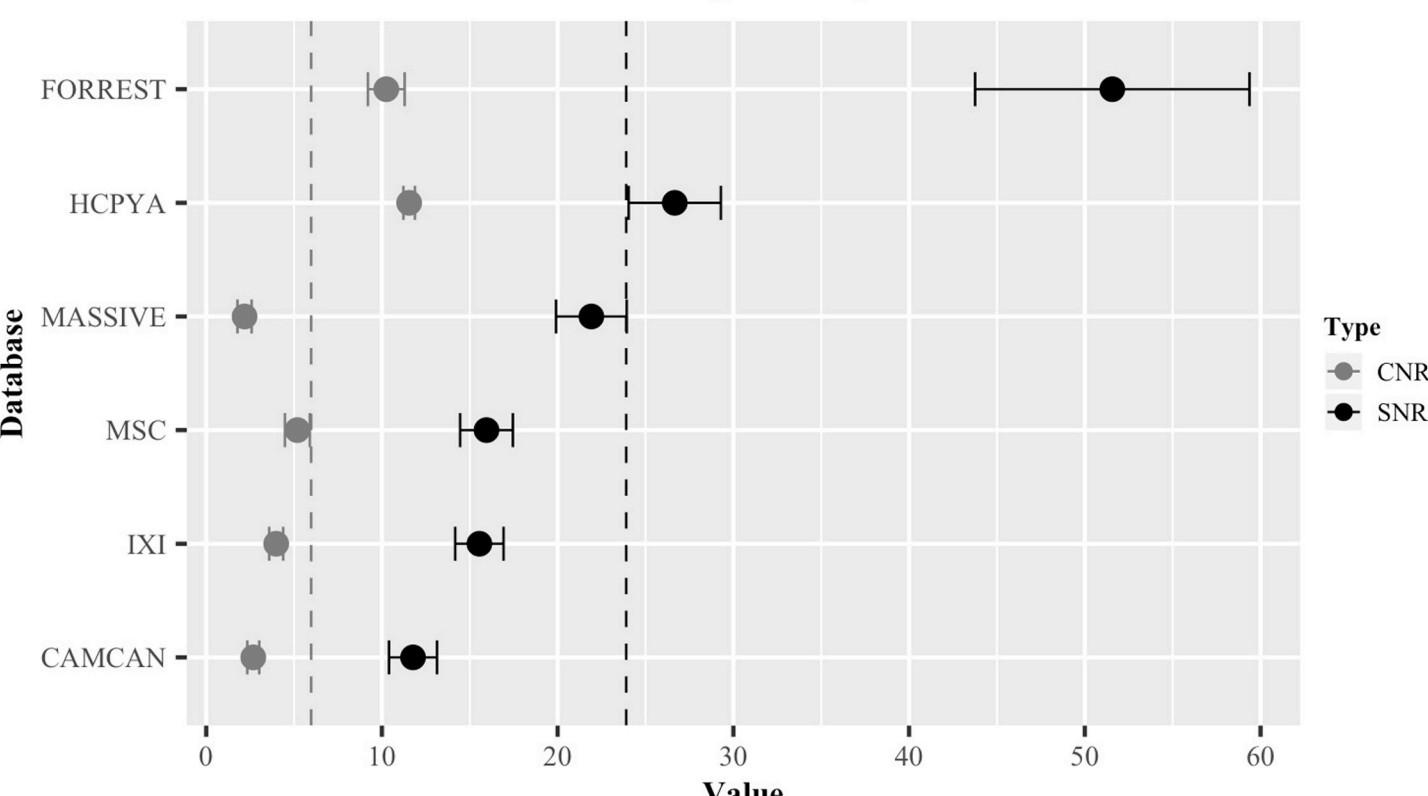

**Fig 7. Overview of the ratios for databases containing a T2w image.** $SNR_{CC}$ values are shown in black, CNR values in grey. Error bars indicate standard error of the mean. The dotted vertical lines indicate the mean of the $SNR_{CC}$ (black) and CNR (grey).

scanning protocols. As only a subset of the databases offered multiple contrasts, direct inter-database comparisons between all contrast types could not be provided. Within this subset, intra-database comparisons between contrasts are possible. SNR and CNR estimations for the contrasts offered by each database are displayed in the S3 Table.

The results of the SNR and CNR calculations show a clear benefit of using UHF MRI, with the five 7T databases (250, AHEAD, ATAG, MAASTRICHT and MPI-CBS) obtaining the largest values in the CC. Moreover, the MPI-CBS and 250 databases showed much higher image quality compared to the other databases. It should be noted, however, that the images offered from the MPI-CBS database include image post-processing pipelines that are not

**Table 3. Summary table describing the $SNR_{CC}$, $SNR_{CN}$ and CNR of the T2w images of each database.**

| Database | Sequence | Contrast | $SNR_{CC}$ | $SNR_{CN}$ | CNR | n |
|---|---|---|---|---|---|---|
| Cam-Can | SPACE | T2w | 11.8 ± 1.4 | 14.3 ± 0.7 | 2.7 ± 0.2 | 15 |
| HCP-YA | SPACE | T2w | 26.7 ± 2.6 | 37.7 ± 3 | 11.5 ± 0.3 | 5 |
| IXI | - | T2w | 15.5 ± 1.4 | 17.7 ± 1 | 4.0 ± 0.2 | 15 |
| MASSIVE | 3DTSE | T2w | 21.9 ± 2.0 | 13.8 ± 1.8 | 2.2 ± 0.4 | 1 |
| MSC | - | T2w | 16.0 ± 1.5 | 23 ± 2.3 | 5.2 ± 0.7 | 5 |
| Forrest | 3DTSE | T2w | 51.6 ± 7.8 | 66.0 ± 3.0 | 10.2 ± 1.0 | 5 |

Each ratio value is shown as the mean of all the subjects ± the standard error of the mean. n indicates the number of subjects used for the calculations.

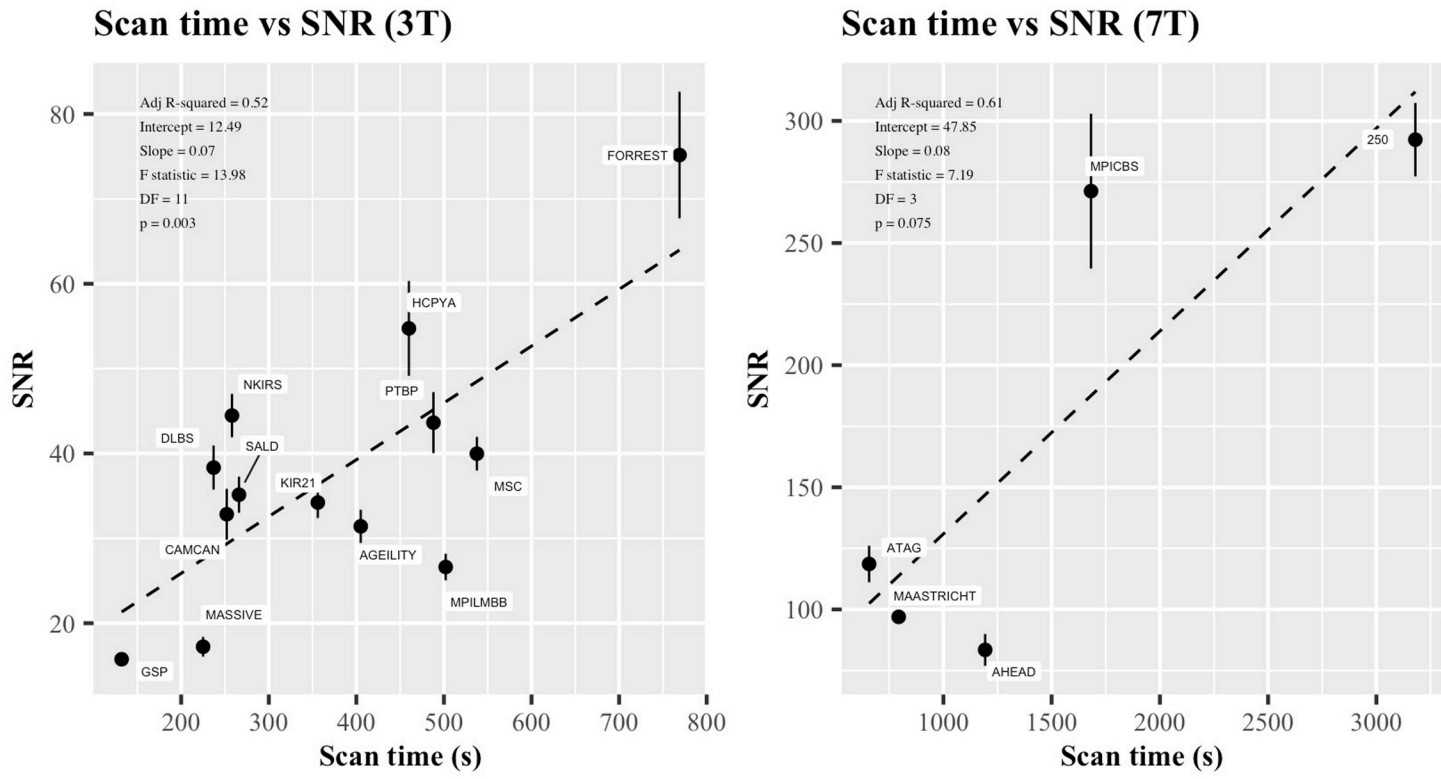

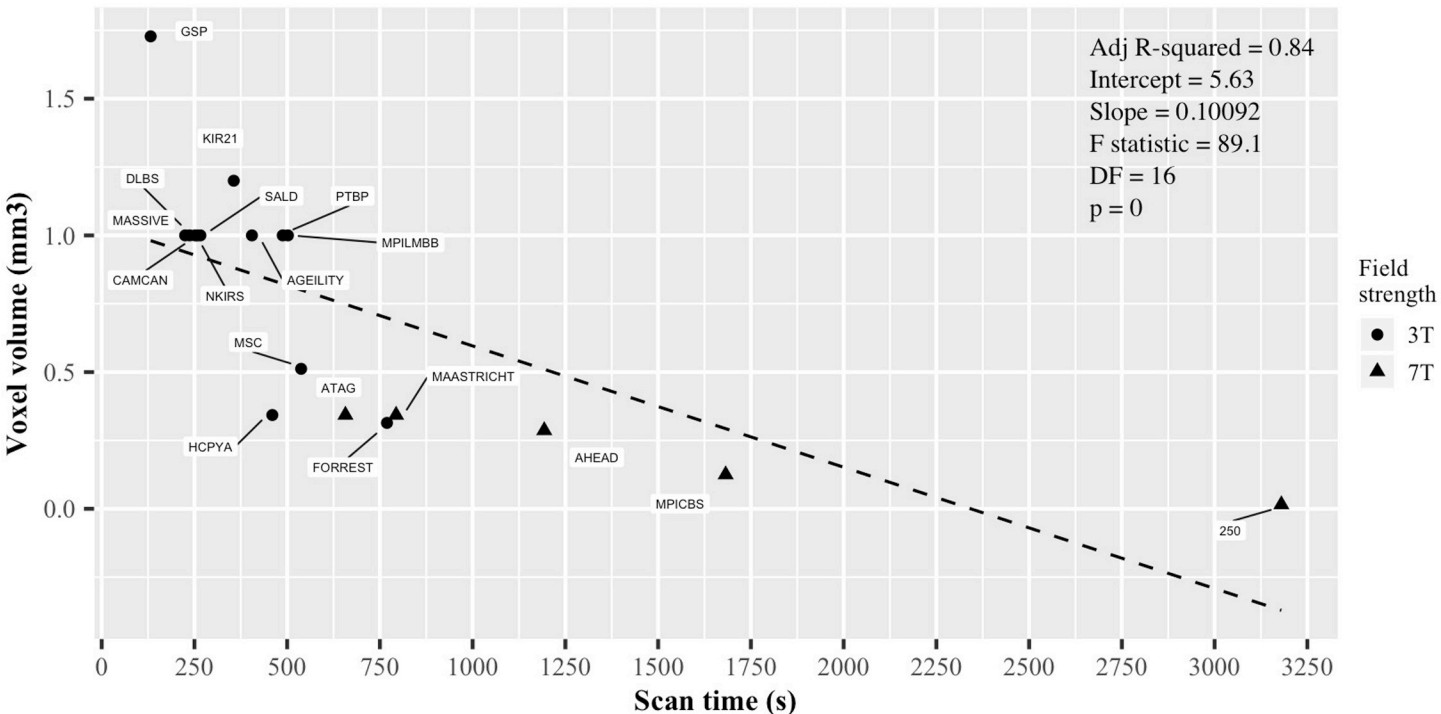

**Fig 8. The relationship between the $SNR_{CC}$ and voxel dimensions of T1w images with scanning time in 18 databases.** A) Graphical representation of $SNR_{CC}$ and scan time. Error bars indicate standard error of the mean. B) Graphical representation of voxel dimensions and scan time. Both legends contain information relating to the adjusted R-squared value, intercept, slope, F statistic, degrees of freedom and p-value. Circular symbols indicate 3T data, triangular symbols indicate 7T data.

## SNR estimations for T1 maps and T1 weighted images

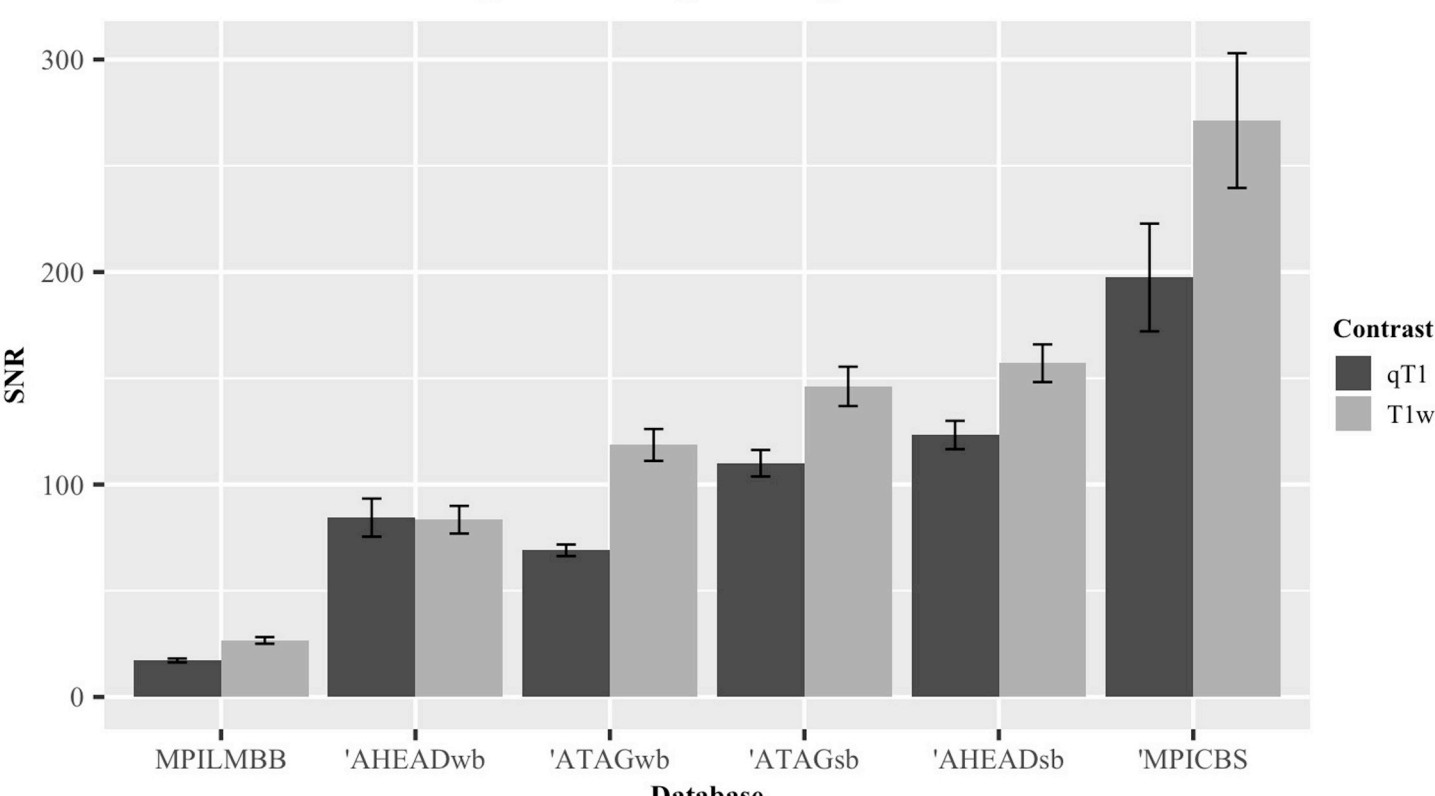

**Fig 9. Graphical representation of difference in normalized SNR$_{CC}$ values for quantitative T1 maps and T1-weighted images.** Databases marked with an apostrophe (') indicate 7T data. qT1, quantitative T1 map; T1w, T1-weighted; wb, whole-brain; sb, slab. Error bars indicate standard error of the mean.

applied in any of the other databases. Such processing pipelines can increase image quality substantially, and are another benefit of openly accessible imaging data and protocols.

**Table 4. Normalized SNR$_{CC}$, SNR$_{CN}$ and CNR values for the databases that presented slabs as well as whole brain data.**

| Database | Sequence | Contrast | Type | SNR$_{CC}$ | SNR$_{CN}$ | CNR |
|---|---|---|---|---|---|---|
| *AHEAD* | MP2RAGEME | qT1 | WB | 84.4 ± 8.3 | 68.4 ± 4.8 | 20.9 ± 2.1 |
| *AHEAD* | MP2RAGEME | qT1 | SB | 123.3 ± 5.8 | 115.0 ± 6.4 | 34.7 ± 1.3 |
| *AHEAD* | MP2RAGEME | T1w | WB | 83.4 ± 6.1 | 39.5 ± 1.5 | 28.5 ± 1.3 |
| *AHEAD* | MP2RAGEME | T1w | SB | 157.1 ± 7.5 | 103.1 ± 5.9 | 37.4 ± 1.3 |
| *AHEAD* | MP2RAGEME | PDw | WB | 97.5 ± 6.2 | 29.5 ± 2.4 | 1.5 ± 0.5 |
| *AHEAD* | MP2RAGEME | PDw | SB | 147.8 ± 8.2 | 79.5 ± 6.4 | 8.3 ± 2.5 |
| *AHEAD* | MP2RAGEME | qT2$^*$ | WB | 37.0 ± 2.1 | 22.1 ± 2.2 | 3.0 ± 0.4 |
| *AHEAD* | MP2RAGEME | qT2$^*$ | SB | 63.8 ± 2.7 | 49.8 ± 4.4 | 6.4 ± 1.9 |
| *ATAG* | MP2RAGE | qT1 | WB | 69.0 ± 2.7 | 51.8 ± 2.4 | 18.0 ± 0.6 |
| *ATAG* | MP2RAGE | qT1 | SB | 110 ± 5.2 | 90.7 ± 3.9 | 22.6 ± 1.2 |
| *ATAG* | MP2RAGE | T1w | WB | 118.6 ± 6.5 | 29.6 ± 1.4 | 19.6 ± 0.7 |
| *ATAG* | MP2RAGE | T1w | SB | 146.2 ± 9.1 | 47.9 ± 2.1 | 23.3 ± 1.3 |

15 subjects were used for all contrast types in these databases. Standard errors of the mean are given for normalized SNR$_{CC}$ and CNR values. qT1, quantitative T1 map; T1w, T1 weighted; PDw, proton density weighted; qT2$^*$, quantitative T2$^*$ map; WB, whole brain; SB, slab.

**SNR estimations of CC for T1w images across age groups**

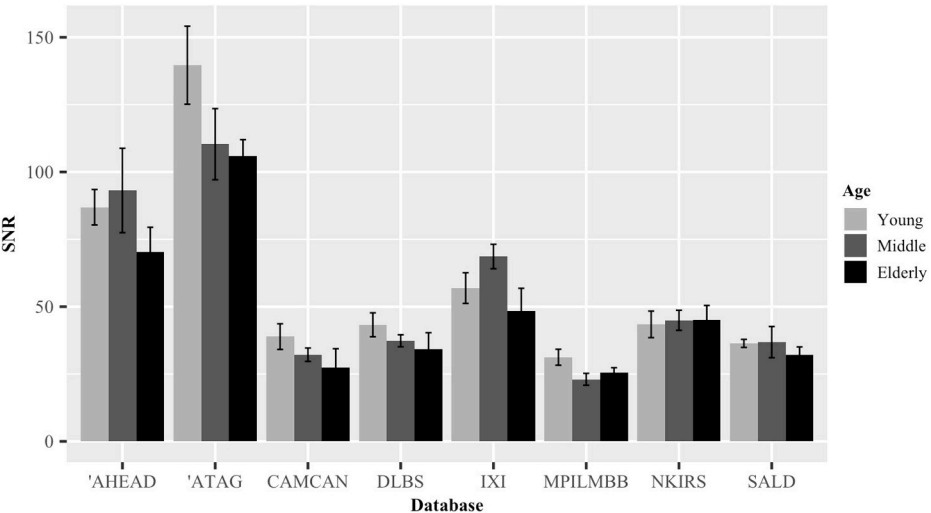

**SNR estimations of CN for T1w images across age groups**

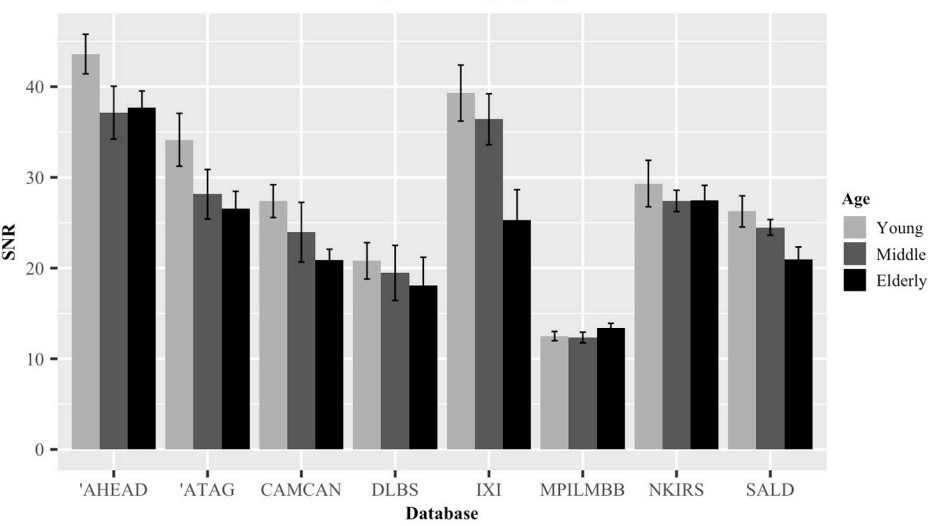

**CNR estimations for T1w images across age groups**

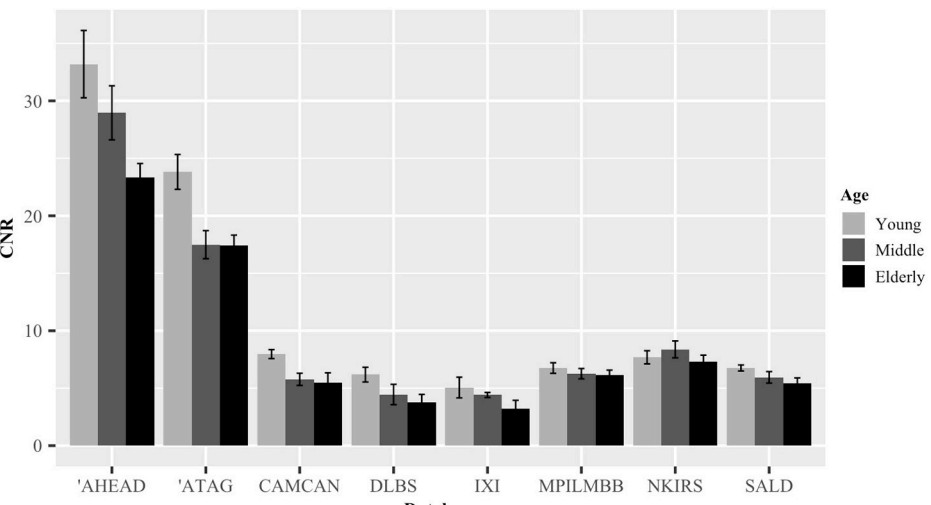

**Fig 10. Comparison of ratios for T1w images across age groups.** A) $SNR_{CC}$. B) $SNR_{CN}$. C) CNR. Error bars indicate standard error of the mean. Each bar singular represents five participants. Databases marked with an apostrophe (') indicate 7T data.

Through the availability of this data, exciting new data pipelines and tools can be developed and shared. In S3 Table, you can find the $SNR_{CC}$, $SNR_{CN}$ and CNRs for all the images analyzed for each database. Note that this includes two databases (HCPYA and 250), which provide both processed and unprocessed images. A clear benefit of post-acquisition processing pipelines can be seen when comparing these ratios within-database. The processed T1w images provided by the 250 database increase the $SNR_{CC}$ from 292.3 ± 15.0 to 570.4 ± 123.5, a similar increase can be seen in the $SNR_{CN}$, increasing from 198.3 ± 14.6 to 368.0 ± 53.5, though this did not benefit the CNR (93.5 ± 6.7 and 93.7 ± 14.6 for the unprocessed and processed images, respectively). Within the HCPYA database, increases in the SNRs and CNRs of both the T1w and T2w images are also apparent. Taken together, this suggests that optimizing post-acquisition processing methods can provide additional increases in image quality that are not trivial.

While the analysis presented here quantifies an important aspect of the databases, they are not the only factor to take into account when selecting imaging data for further research purposes. At an overview, our results indicate a strong advocation for the MPI-CBS and 250 databases, owing to their SNR and CNR far above the rest. However, there are also other factors to consider, for example, sample size, age-range and the number of contrasts included are just some of a long list of criteria many research questions need to consider. As such, the small sample size, limited age-range and limited contrasts make the MPI-CBS and 250 databases less attractive for many lines of research.

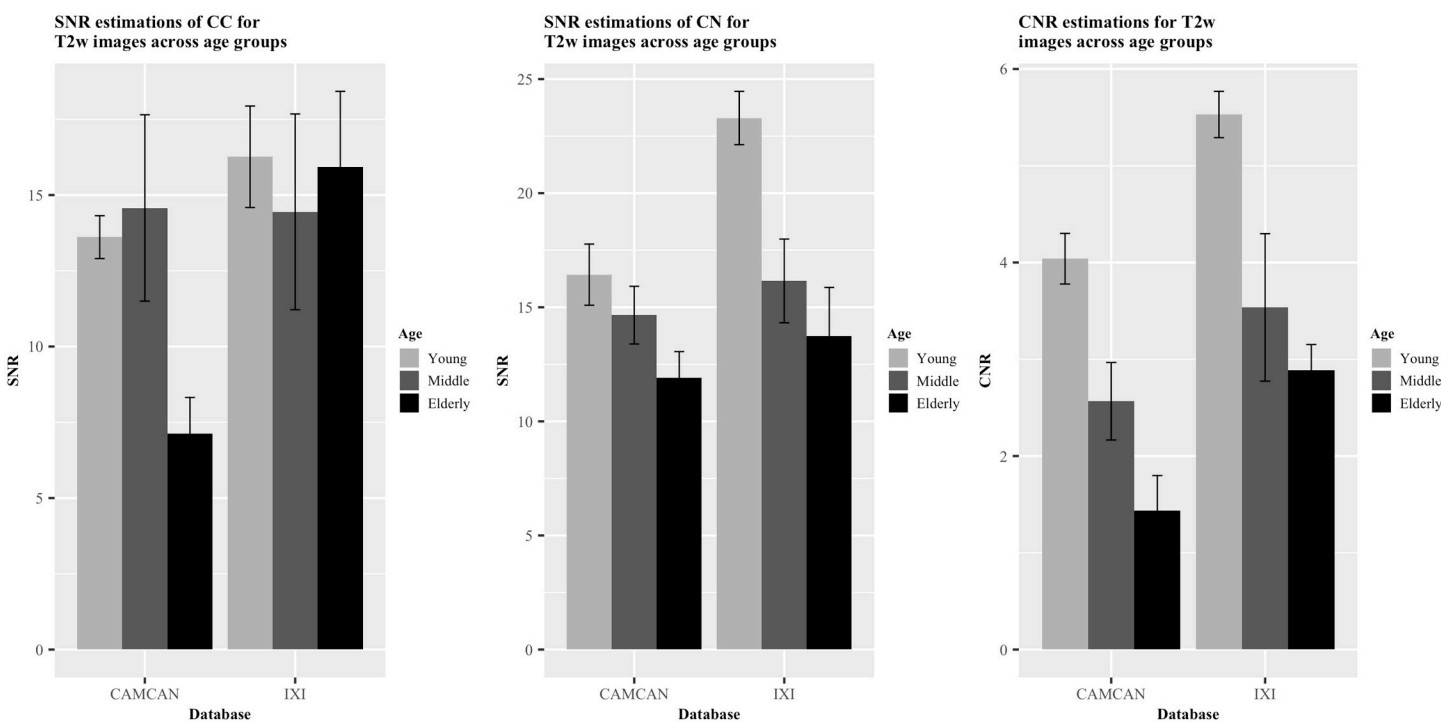

**Fig 11. Comparison of ratios for T2w images across age groups.** A) $SNR_{CC}$. B) $SNR_{CN}$. C) CNR. Error bars indicate standard error of the mean. Each bar singular represents five participants.

Although the relationships of SNR and voxel dimensions with scan time presented here are obvious and reflect basic MRI physics, it is nonetheless interesting to see the efficiency of separate MRI protocols. These relationships are particularly informing when aspects of image quality largely deviate from linearity. Optimized MR sequences or contrasts that allow for high spatial resolution or high SNRs with short scan times offer preferable performance. These comparisons also indicate further favourability for 7T imaging, with most of the resultant 7T database images residing on the efficient side of the linear trends displayed with scan time.

We present here age-related differences across four MRI contrasts; T1w, T2w, qT1, and qT2*. Our SNR-CN analyses suggest a consistent age-related decline in all image types. Age-related changes in relaxation values in the human brain have been long-known [101]. There is also evidence that T2* values reflect iron concentration in neural tissues [102]. During healthy aging, iron-deposition appears to increase in some brain structures (e.g., parts of the basal ganglia) [103–106]. The decline of qT2* measures in the CN therefore likely reflects this increase in iron deposition. The lowering in effective T2 found here is also in line with previous work [107, 108]. Volume loss in this region is another known process observed in the healthy aging brain [109, 110], which would be accompanied by a declining proton density, lowering the signal derived from T1 recovery and T2 relaxation. Taken together, these declines in signal would suggest an age-related structural change of the caudate nuclei.

A more complicated picture is painted for the SNR$_{CC}$ measurements. A SNR decline in the CC was found in T1w, qT1 and qT2* images, though this decline is not as apparent as in the CN. *Post mortem* histological analyses of white matter regions have shown that the myelination of nerve fibres decreases with age [111]. This process of demyelination is associated with an increase in SNR in qT1 and T1w images [108], in opposition to what was found here. It should be noted, however, that there appears to be an increase in image noise in the elderly population (measured as the standard deviation of the 27 voxels measured per image). This increase in noise was not accompanied by a decrease in mean signal of the region, and therefore likely drives the small decline in SNR found. For the other relationships, this increase in noise as a function of age is also apparent. However, since this increase is also accompanied by a decrease in mean signal, it most likely reflects an underlying structural change. We note that age-related structural changes are heterogeneous across different regions of the brain. The processes underlying these changes are similarly heterogeneous and a combination of a multitude of factors, including changes in the small vessels supplying the regions, regional brain atrophy, loss of myelination and impaired white matter [111–114]. These changes, in addition to increased subject motion during scanning could all impact the increase level of noise found in the elderly population. It has been suggested that head motion increases as a function of age [115], although some findings have suggested a more non-linear relationship between the two [116]. Even subtle forms of motion artefacts have been shown to affect interpretability of imaging analysis results (e.g., cortical thickness estimates; [117]). Image noise introduced through head motion also lowers SNR estimates and degrades image quality [118]. This highlights the need for motion correction in structural MRI. Due to our limited snapshot of the data available, we can only show results that hint at these intricate relationships.

For the CNR measurements, there consistently appears to be an age-related decline across the adult lifespan, as indicated by the analysis of all four contrasts. Such CNR differences are also found when comparing adult and infant brains [119]. This decrease in CNR over the adult lifespan is a by-product of the physical changes to the contrasted regions (CC and CN). The observed decrease in SNR in these two regions leads to this decrease in CNR. The analyses of age-related differences presented here illustrates just one of the many interesting ways these open-access databases can be used for in the future.

It should be stressed that there are a variety of methods to calculate both the SNR and CNR of structural MR images, note that most of these methods are not applicable to all situations. For SNR estimation, there are two other prominent methods used in the field. The first involves measuring the SNR of the region of interest (ROI) within the brain and dividing it by the SNR of the background of the image outside of the brain. The second involves measuring only the mean signal of the ROI inside the brain and dividing it by the standard deviation of a region outside of the brain. A commonality in both of these methods is that they assume that measuring an area outside of the brain captures only the noise induced by the MR scanner itself. One reason we opted for the method used here is that due to inhomogeneities in the magnetic field of each scanner and differences in the spatial distribution of noise [120, 121], the area of the background image chosen for the measurement of noise could differ significantly between sites and sequences. Of course, our method does not remove the problem of bias, but as this bias is the same across all of the images measured here, we believe the comparison is fair. Regardless of the method used for the measurement of the SNR, the most important requirement for an objective comparison is that the method used is consistent across all data. To signify that this method was indeed reliable within the databases, we ran the validation study on the separate T1w images. The reproducibility of the estimates that we took indicate that the methods holds as a consistent measurement of SNR.

As spatial resolution increases, sensitivity to both voluntary or involuntary motion and physiological noise will also increase, and therefore continue to be a ceiling on image quality at all field strengths. Methods to overcome such movement artefacts include both retrospective and prospective motion correction [122, 123]. Both approaches have displayed their ability to increase image quality at 3T and 7T, providing a way around subject motion at high resolution [38, 124–126]. Removing this confound completely while scanning healthy individuals is infeasible, but *post mortem* MRI can benefit from the lack of movement artefacts, allowing for scan times inconceivable in live subjects. These scan times can facilitate the visualization of a much larger number of smaller brain structures [127]. For the purpose of creating probabilistic atlases of the human brain, such a technique when used in concurrence with histological methods can provide greater detail than *in vivo* MRI alone [29].

We acknowledge that for many of the databases discussed here, we have only analyzed a snapshot of the data and have not taken advantage of all of the data we have access to. This limitation was necessary to keep our analysis level feasible, as the range in sizes of these databases make using all participants problematic. For the future, we would hope that a standardized SNR protocol will become a feature that all new databases will use and present with their data. Ideally, this would include manually segmented masks of the same anatomical areas, from unprocessed images in their native spaces. We also hope that open-access databases continue to become the norm across the scientific field.

## Conclusion

The current study provides a quantitative comparison between some of the most fruitful open-access neuroimaging databases available, which can aid researchers in selecting which databases to use. The results presented here give an indication of the large variation in image quality provided by these databases. The estimations (SNR and CNR), as well as the number of contracts provided by each database (as these give visual information to specific tissue types), can aid in the selection process. The benefit of large-scale imaging databases for creating general maps of cortical organization and providing both phenotypic and genetic comparisons across populations is clear. However, large-scale databases often come at the cost of lower image resolution due to the financial implications of using large sample

sizes, ultra-high field MRI and extensive scan times. In particular for the human subcortex, image resolution is critical and standard structural 3T MRI data does not provide the required resolution and SNR for small nuclei. The higher quality of 7T databases provides a clear advantage, but high cost and limited access are still preventing the collection of larger cohorts. Each database presented here has assisted an important neuroscientific movement towards open-access imaging data. With the number of subjects ranging from one to over 1500 and the number of sessions from one to 18, the objectives and characteristics of these databases are diverse. We hope that our current efforts will help researchers to choose the appropriate database for their research question and highlight their usefulness to the scientific field in the study of normative human brain structure.

## Supporting information

**S1 Table. Websites, descriptor papers and download process for all presented databases.** N, no; Y, yes; U, unprocessed; P, processed; N/A, not applicable; NITRC, Neuroimaging Tools & Resources Collaboratory.
(DOCX)

**S2 Table. High-profile databases that do not adhere to all of our inclusion criteria.** ABCD; Adolescent Brain Cognitive Development study, ABIDE; Autism Brain Imaging Data Exchange, ADHD-200; Attention Deficit Hyperactivity Disorder, ADNI; Alzheimer's Disease Neuroimaging Initiative, AIBL; Australian Imaging Biomarkers and Lifestyle Study of Aging, BRAINS; Brain Images of Normal Subjects, CMI-HBN; Child Mind Institute Healthy Brain Network, COBRE; Center for Biomedical Research Excellence, CoRR; Consortium for Reliability and Reproducibility, fBIRN; Function Biomedical Informatics Research Network, FCP; 1000 Functional Connectome Project, MIRIAD; Minimal Interval Resonance Imaging in Alzheimer's Disease, NACC; National Alzheimer's Coordinating Center, NCANDA; National Consortium on Alcohol and Neurodevelopment in Adolescence, OASIS; Open Access Series of Imaging Studies, PING; Pediatric Imaging, Neurocognition, and Genetics, PNC; Philadelphia Neurodevelopmental Cohort, UK; United Kingdom. + indicates that this database meets this inclusion criteria,—indicates that this database does not meet this inclusion criteria.
(DOCX)

**S3 Table. SNR and CNR estimations for the contrasts offered by each database.** All qualitative images are reported as normalized values, all quantitative images are reported as raw. $SNR_{CC}$ indicates the SNR of the corpus callosum and $SNR_{CN}$ indicates the mean SNR of both caudate nuclei. n, number of participants used for calculation; MPRAGE, magnetization prepared rapid gradient echo; MP2RAGE, magnetization prepared 2 rapid acquisition gradient echoes; ME, multiple echo; FLASH, fast low angle shot; SPACE, sampling perfection with application of optimized contrasts using different flip angle evolutions; FLAIR, fluid attenuation inversion recovery; IR, inversion recovery; TSE, turbo spin echo; TFE, turbo field echo; sb, slab; un, unprocessed; pro, processed; T1w, T1-weighted; qT1, quantitative T1 map; T2w, T2-weighted; PDw, proton density-weighted; $T2^*w$, $T2^*$-weighted; $qT2^*$, quantitative $T2^*$ map; QSM, quantitative susceptibility mapping; SWI, susceptibility weighted image.
(DOCX)

**S1 Fig. Bar chart comparing the raw and normalized SNR measurements.** Values are ordered from lowest to highest, based on the raw SNR measurements. Error bars indicate standard error of the mean.
(TIFF)

**S1 Checklist. PRISMA 2009 checklist.**
(DOC)

# Acknowledgments

Data were provided [in part] by the Brain Genomics Superstruct Project of Harvard University and the Massachusetts General Hospital, (Principal Investigators: Randy Buckner, Joshua Roffman, and Jordan Smoller), with support from the Center for Brain Science Neuroinformatics Research Group, the Athinoula A. Martinos Center for Biomedical Imaging, and the Center for Human Genetic Research. 20 individual investigators at Harvard and MGH generously contributed data to the overall project. Data used in the preparation of this work were obtained from the CamCAN repository (available at http://www.mrc-cbu.cam.ac.uk/datasets/camcan/) [64, 65]. Data collection and sharing for this project was provided by the Cambridge Centre for Ageing and Neuroscience (CamCAN). Data were provided [in part] by the Human Connectome Project, WU-Minn Consortium (Principal Investigators: David Van Essen and Kamil Ugurbil; 1U54MH091657). The Midnight Scan Club data was obtained from the OpenfMRI database. Its accession number is ds000224. The data from Leipzig Mind Brain Body was obtained from the OpenfMRI database. Its accession number is ds000221.

# Author Contributions

**Conceptualization:** Scott Jie Shen Isherwood, Pierre-Louis Bazin, Anneke Alkemade, Birte Uta Forstmann.

**Data curation:** Scott Jie Shen Isherwood.

**Formal analysis:** Scott Jie Shen Isherwood.

**Investigation:** Scott Jie Shen Isherwood, Pierre-Louis Bazin, Birte Uta Forstmann.

**Methodology:** Scott Jie Shen Isherwood, Pierre-Louis Bazin, Anneke Alkemade, Birte Uta Forstmann.

**Supervision:** Pierre-Louis Bazin, Birte Uta Forstmann.

**Writing – original draft:** Scott Jie Shen Isherwood.

**Writing – review & editing:** Scott Jie Shen Isherwood, Pierre-Louis Bazin, Anneke Alkemade, Birte Uta Forstmann.

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
