## [Decision Letter · Decision Letter 0]

24 Sep 2020

PONE-D-20-17959

Quantity and quality: Normative open-access neuroimaging databases

PLOS ONE

Dear Dr. Isherwood,

Thank you for submitting your manuscript to PLOS ONE. After careful consideration, we feel that it has merit but does not fully meet PLOS ONE’s publication criteria as it currently stands. Therefore, we invite you to submit a revised version of the manuscript that addresses the points raised during the review process.

We look forward to receiving your revised manuscript.

Kind regards,

Zhaolin Chen

Academic Editor

PLOS ONE

Journal Requirements:

'AHEAD data was collected with financial support from NWO-Vici (BUF) and NWO-STW (AA and BUF).

'The author(s) received no specific funding for this work.'

Additional Editor Comments:

Dear authors,

Apologise for the delay in processing your paper. We have tried to reach one review for their comments however the reviewer is not responding to our requests over the past month or so. To proceed your paper, I have one reviewer's report, and I have read the paper my self and would like to provide two comments: (i) I recommend you to clearly state the need for SNR and CNR metrics in these datasets. You mentioned briefly in the introduction section that cortical surface mapping is dependent on SNR and CNR, however I think you need to cite references and review thoroughly any analysis bias has been induced due to SNR and CNR variations in the literature. (ii) there are many ways to calculate SNR and CNR, I recommend your discuss this in your discussion section.

Reviewers' comments:

Reviewer's Responses to Questions

**Comments to the Author**

1. Is the manuscript technically sound, and do the data support the conclusions?

Reviewer #1: Partly

2. Has the statistical analysis been performed appropriately and rigorously? 

Reviewer #1: No

3. Have the authors made all data underlying the findings in their manuscript fully available?

Reviewer #1: Yes

4. Is the manuscript presented in an intelligible fashion and written in standard English?

Reviewer #1: Yes

5. Review Comments to the Author

Reviewer #1: The authors have presented an analysis on the SNR and CNR for 20 publically available MRI databases and also compared the age related changes in SNR and CNR. The manuscript is organized well and is informative. Following are some concerns/comments:

Major:

1. Some of the databases have large number of images, and thus taking 5 sample randomly may not be a true reflection of the whole population. Can authors randomly take separate 5 samples from each database (where available) and check if the same analysis still holds?

2. It appears that the SNR was calculated using 3x3x3 voxel only? If only a single ROI of 3x3x3 was used for calculation then it is prone to large errors due to a few pixels offset in the location of the ROI. A better estimate might be to perform a moving window averaging of 3x3x3 mask over a larger ROI (CC, CN)

Minor:

1. Fig. 1, please indicate CC, RCN, LCN regions PRISMAin the images.

2. PRISMA abbreviation?

3. It might be better to perform separate analysis of scan time vs SNR for 3T and 7T. Since SNR depends on both the field strength and scan time.

4. Which data supports the following line in the conclusion.

“The benefit of large-scale imaging databases for creating general maps of cortical organization and providing both phenotypic and genetic comparisons across populations is clear.”

5. Another line in conclusion “In particular for the human subcortex ….. does not provide the required resolution and SNR for small nuclei.” There was no analysis on the resolution in this study?

6. PLOS authors have the option to publish the peer review history of their article (what does this mean?). If published, this will include your full peer review and any attached files.

Reviewer #1: No

---

## [Author Response · Author response to Decision Letter 0]

18 Jan 2021

We thank the editor and reviewers for their comments. Our point-by-point responses to these comments can be found in the 'response_to_reviewers_plosone_QQ' document.

---

## [Decision Letter · Decision Letter 1]

25 Feb 2021

Quantity and quality: Normative open-access neuroimaging databases

PONE-D-20-17959R1

Dear Dr. Isherwood,

We’re pleased to inform you that your manuscript has been judged scientifically suitable for publication and will be formally accepted for publication once it meets all outstanding technical requirements.

Kind regards,

Zhaolin Chen

Academic Editor

PLOS ONE

Additional Editor Comments (optional):

Reviewers' comments:

Reviewer's Responses to Questions

**Comments to the Author**

1. If the authors have adequately addressed your comments raised in a previous round of review and you feel that this manuscript is now acceptable for publication, you may indicate that here to bypass the “Comments to the Author” section, enter your conflict of interest statement in the “Confidential to Editor” section, and submit your "Accept" recommendation.

Reviewer #1: All comments have been addressed

2. Is the manuscript technically sound, and do the data support the conclusions?

Reviewer #1: Yes

3. Has the statistical analysis been performed appropriately and rigorously? 

Reviewer #1: Yes

4. Have the authors made all data underlying the findings in their manuscript fully available?

Reviewer #1: Yes

5. Is the manuscript presented in an intelligible fashion and written in standard English?

Reviewer #1: Yes

6. Review Comments to the Author

Reviewer #1: (No Response)

7. PLOS authors have the option to publish the peer review history of their article (what does this mean?). If published, this will include your full peer review and any attached files.

Reviewer #1: No

---

## [Editor Report · Acceptance letter]

2 Mar 2021

PONE-D-20-17959R1 

Quantity and Quality: Normative Open-access Neuroimaging Databases 

Dear Dr. Isherwood:

I'm pleased to inform you that your manuscript has been deemed suitable for publication in PLOS ONE. Congratulations! Your manuscript is now with our production department. 

Kind regards, 

on behalf of

Dr. Zhaolin Chen 

Academic Editor

PLOS ONE